# Rift Valley fever phlebovirus NSs protein core domain structure suggests molecular basis for nuclear filaments

Michal Barski[1†‡], Benjamin Brennan[2†], Ona K Miller[1], Jane A Potter[1], Swetha Vijayakrishnan[2], David Bhella[2], James H Naismith[1], Richard M Elliott[2§], Ulrich Schwarz-Linek[1]*

[1]Biomedical Sciences Research Complex, University of St Andrews, St Andrews, United Kingdom; [2]MRC-University of Glasgow Centre for Virus Research, London, United Kingdom

*For correspondence:
us6@st-andrews.ac.uk

[†]These authors contributed equally to this work

Present address: [‡]Division of Infectious Diseases, Imperial College London, London, United kingdom

[§]Deceased

Competing interests: The authors declare that no competing interests exist.

**Abstract** Rift Valley fever phlebovirus (RVFV) is a clinically and economically important pathogen increasingly likely to cause widespread epidemics. RVFV virulence depends on the interferon antagonist non-structural protein (NSs), which remains poorly characterized. We identified a stable core domain of RVFV NSs (residues 83–248), and solved its crystal structure, a novel all-helical fold organized into highly ordered fibrils. A hallmark of RVFV pathology is NSs filament formation in infected cell nuclei. Recombinant virus encoding the NSs core domain induced intranuclear filaments, suggesting it contains all essential determinants for nuclear translocation and filament formation. Mutations of key crystal fibril interface residues in viruses encoding full-length NSs completely abrogated intranuclear filament formation in infected cells. We propose the fibrillar arrangement of the NSs core domain in crystals reveals the molecular basis of assembly of this key virulence factor in cell nuclei. Our findings have important implications for fundamental understanding of RVFV virulence.

DOI: https://doi.org/10.7554/eLife.29236.001

## Introduction

Rift Valley fever phlebovirus (RVFV), of the genus *Phlebovirus*, is one of the most clinically significant members of the *Phenuiviridae* family, of the *Bunyavirales* order (*Elliott and Brennan, 2014*; *Plyusnin et al., 2012*; *Adams et al., 2017*). RVFV is an arbovirus spread by many mosquito vector species as well as by exposure to infected tissues. It causes recurring epidemics in livestock and humans (*Ikegami and Makino, 2011*). The most prominent effect of RVFV infection in ruminants is a high rate of abortions. In humans infections are usually self-limiting, but can develop into hepatitis, retinitis, encephalitis and hemorrhagic fever with fatality rates of 0.5–2% (*Pepin et al., 2010*). Although originally endemic to sub-Saharan Africa, RVFV has recently appeared in Madagascar, the Comoros and the Arabian Peninsula (*Balkhy and Memish, 2003*). Increasing spread of competent mosquito vector species due to climate change could facilitate emergence of this virus in new eco-systems, including Europe and the United States (*Chevalier, 2013*; *Elliott, 2009*; *Golnar et al., 2014*; *Rolin et al., 2013*). In 2017 the World Health Organization ranked RVFV among the ten most dangerous pathogens most likely to cause wide epidemics in the near future, requiring urgent attention (http://www.who.int/blueprint/priority-diseases/en/). While an animal vaccine for RVFV exists, there is no treatment available for human use (*Boshra et al., 2011*; *Lihoradova and Ikegami, 2014*).

RVFV is a zoonotic arbovirus evolved to evade the immune response of mammals and insects. Innate immune antagonism of RVFV is primarily mediated by its main virulence factor, the 30 kDa non-structural protein (NSs), which is transcribed and translated from a subgenomic mRNA derived

**eLife digest** Rift Valley fever phlebovirus (RVFV) is a virus of humans and livestock, transmitted by mosquitos and contact with infected animals. Infection can cause severe disease, including hemorrhagic fever, and may lead to death. Historically, the virus was only found in central Africa but it has spread for instance to the Arabian Peninsula. There is a risk that the virus may appear in temperate regions including Europe because global warming is allowing the mosquitos that carry the virus to extend their geographic range. There are no vaccines or treatments available for use in humans so if there is a serious outbreak of the virus it could become an epidemic and cause great economic losses and severe human disease.

RVFV relies on a protein called NSs to cause disease. In cells of infected animals and humans NSs forms filaments inside the nucleus, the control center of the cell, and disarms the immune system. However, it is not known precisely how NSs works.

To address this question, Barski, Brennan et al. used a technique called X-ray crystallography to study the atomic three-dimensional structure of NSs. This revealed that the center of the protein contains a core domain that causes the filaments to form. Further experiments identified how the NSs core comes together to build the filaments inside the cell nucleus.

These findings represent an important step towards understanding how the NSs protein helps RVFV to cause disease in humans and livestock. In the future, this work may aid the development of much needed drugs and vaccines against RVFV.

DOI: https://doi.org/10.7554/eLife.29236.002

from the antigenomic S segment early during infection. NSs impedes interferon production through three known mechanisms (*Billecocq et al., 2004*; *Bouloy et al., 2001*). In infected host cells, NSs is transported into the nucleus where it interferes with the assembly of the RNA polymerase II preinitiation complex transcription factor II H (TFIIH) by binding to the p44 subunit (*Le May et al., 2004*) and directing the p62 subunit for degradation (*Kainulainen et al., 2014*; *Kalveram et al., 2011*), thereby halting global cellular transcription. NSs also prevents activation of the interferon-β promoter specifically by binding to factor SAP30, a member of a multisubunit histone deacetylase transcriptional repression complex that regulates interferon-β expression (*Le May et al., 2008*). Furthermore, NSs blocks host recognition of viral dsRNA by targeting the RNA-dependent protein kinase (PKR) for degradation (*Ikegami et al., 2009a*, *2009b*). RVFV with a deletion of NSs cannot establish viremia and is not pathogenic in the mouse model (*Bouloy et al., 2001*).

A characteristic feature of RVFV pathogenesis is formation of 0.5 μm thick proteinaceous filaments formed by bundles of thin fibrils in the nuclei of infected cells (*Swanepoel and Blackburn, 1977*). The filaments are composed primarily of NSs protein (*Struthers and Swanepoel, 1982*; *Yadani et al., 1999*) but co-localize with p44, XPB, SAP30 and YY1, as well (*Le May et al., 2004*, *2008*). It is not clear if filament formation is required for NSs function, that is if any interaction sites are present on the monomeric form of the protein, or are only present in the multimerized, fibrillar NSs. In the absence of structural information it is difficult to dissect NSs function from filament formation. Potential functions of nuclear NSs filaments include sequestering NSs binding partners, causing cell cycle arrest and defects in chromosome cohesion and segregation (*Mansuroglu et al., 2010*). This could contribute to the high rate of fetal deformities and abortions observed in RVFV-infected ruminants (*Baer et al., 2012*; *Mansuroglu et al., 2010*).

NSs proteins are encoded in the S segment of the tripartite negative sense single strand RNA genome of most members of the *Phlebovirus* genus, and other families of the *Bunyavirales* order such as the *Peribunyaviridae*, *Nairoviridae*, and some members of the *Hantaviridae* (*Ly and Ikegami, 2016*). They constitute notoriously difficult targets for structural characterization due to an inherent tendency to multimerize, as well as the putative presence of substantial intrinsically unfolded regions. While high-resolution structures of glycoproteins (*Dessau and Modis, 2013*) and nucleoproteins (*Ferron et al., 2011*; *Raymond et al., 2010*) from RVFV and several other bunyaviruses have been determined, no structural information is currently available for any NSs protein, significantly limiting the current understanding of key bunyaviral virulence mechanisms. Structure and function can also not be predicted based on sequence, as NSs proteins are 'ORFans', with very little

sequence similarity in members of the *Bunyavirales* order and the *Phlebovirus* genus. Despite large sequence diversity, NSs proteins examined thus far have been found to play roles in the suppression of the host innate immune response. However, this and the diverse mechanisms of NSs function have been studied only for a few viruses.

Here we describe the first high-resolution structure of a bunyaviral NSs protein. A truncated stable, soluble core domain spanning residues 83–248 of RVFV NSs (NSs-ΔNΔC) was identified using NMR spectroscopy. The corresponding crystal structure represents a novel all-helical fold. Notably, NSs molecules packed in crystals in double-helical fibrils stabilized by multiple interfaces. Dimensions of these fibrils are in good agreement with fibrils constituting nuclear NSs filaments in RVFV-infected cells. Cells infected with recombinant RVFV encoding NSs-ΔNΔC produced nuclear filaments indistinguishable from filaments observed in cells infected with parental RVFV. Furthermore, recombinant viruses encoding full-length NSs with mutations predicted to destabilize significant fibril interfaces, as suggested by crystal packing, did not induce nuclear filaments. Combination of structural biology with cell infection assays using reverse genetics to tailor the main RVFV virulence factor identified a RVFV NSs core domain sufficient for filament formation in vivo, and revealed the structural basis for NSs nuclear filament formation.

## Results

### NSs contains a stable central core and a disordered C-terminal tail

Full-length NSs protein (residues 1–265) was expressed as a cleavable MBP fusion yielding only a small amount (about 0.1 mg per litre of culture) of soluble protein after removal of MBP. Circular dichroism spectroscopy showed secondary and tertiary structure in mature NSs (*Figure 1—figure supplement 1*), however the protein formed large soluble aggregates, eluting in the void volume of size exclusion chromatography (mass over 200 kDa) (*Figure 1A*). This construct was therefore redesigned in order to obtain a stable, non-aggregating and soluble fragment of NSs. Secondary structure predictions suggested an N-terminal region with a propensity to form β sheet structure (residues 1–70), connected by an unstructured linker to a central α-helical domain (85-230). The C-terminal region was predicted to be natively unfolded (231-265). Deletion of the N-terminal 82 amino acids (resulting in truncated NSs hereafter referred to as NSs-ΔN, residues 83–265) significantly improved expression yields, solubility and stability. In solution, the NSs-ΔN protein existed in equilibrium between monomeric and multimeric forms (*Figure 1A*). Near-UV CD spectra indicated a helical protein, with characteristic minima at 208 and 220 nm (*Figure 1—figure supplement 1*).

NSs-ΔN was highly soluble (up to 30 mg/mL) in near-physiological buffer conditions. A 2D $^1$H-$^{15}$N heteronuclear single-quantum coherence (HSQC) NMR spectrum showed well-dispersed crosspeaks, indicative of folded, monomeric protein (*Figure 1B*). The central region of the spectrum, spanning chemical shifts of 7.5 to 8.5 ppm in the $^1$H dimension, was dominated by 17 intense and narrow backbone amide signals, indicating significant flexibility of corresponding residues. Based on secondary structure and disorder predictions, we concluded the intrinsically unfolded residues were present at the C-terminus of NSs-ΔN. Subsequent removal of the C-terminal 17 amino acids (resulting in doubly truncated NSs hereafter referred to as NSs-ΔNΔC, residues 83–248) yielded protein present as a monomer in solution (*Figure 1A*). Its $^1$H-$^{15}$N HSQC spectrum was devoid of all but two intense crosspeaks observed for NSs-ΔN (*Figure 1B*). All other signals overlapped well with crosspeaks in the NSs-ΔN spectrum, indicating absence of residues 249–265 does not influence the structure of NSs-ΔNΔC.

### Crystal structure shows the NSs core domain adopts a novel fold

The NSs-ΔNΔC protein was crystallized at conditions optimized following routine sparse-matrix crystallization screens. The crystals were found to be extremely fragile, and diffracted synchrotron X-rays to 2.2 Å resolution. Crystals belonged to space group *P*6₄22. Calculation of the Matthews coefficient indicated a high solvent content of 76%, assuming two molecules in the asymmetric unit. Due to the absence of any molecular replacement models, experimental phasing had to be undertaken. Because of the higher-than-average sulfur content of NSs-ΔNΔC (12 sulfurs per chain of 175 residues) and the highly symmetrical *P*6₄22 space group of the crystals, S-SAD phasing was pursued.

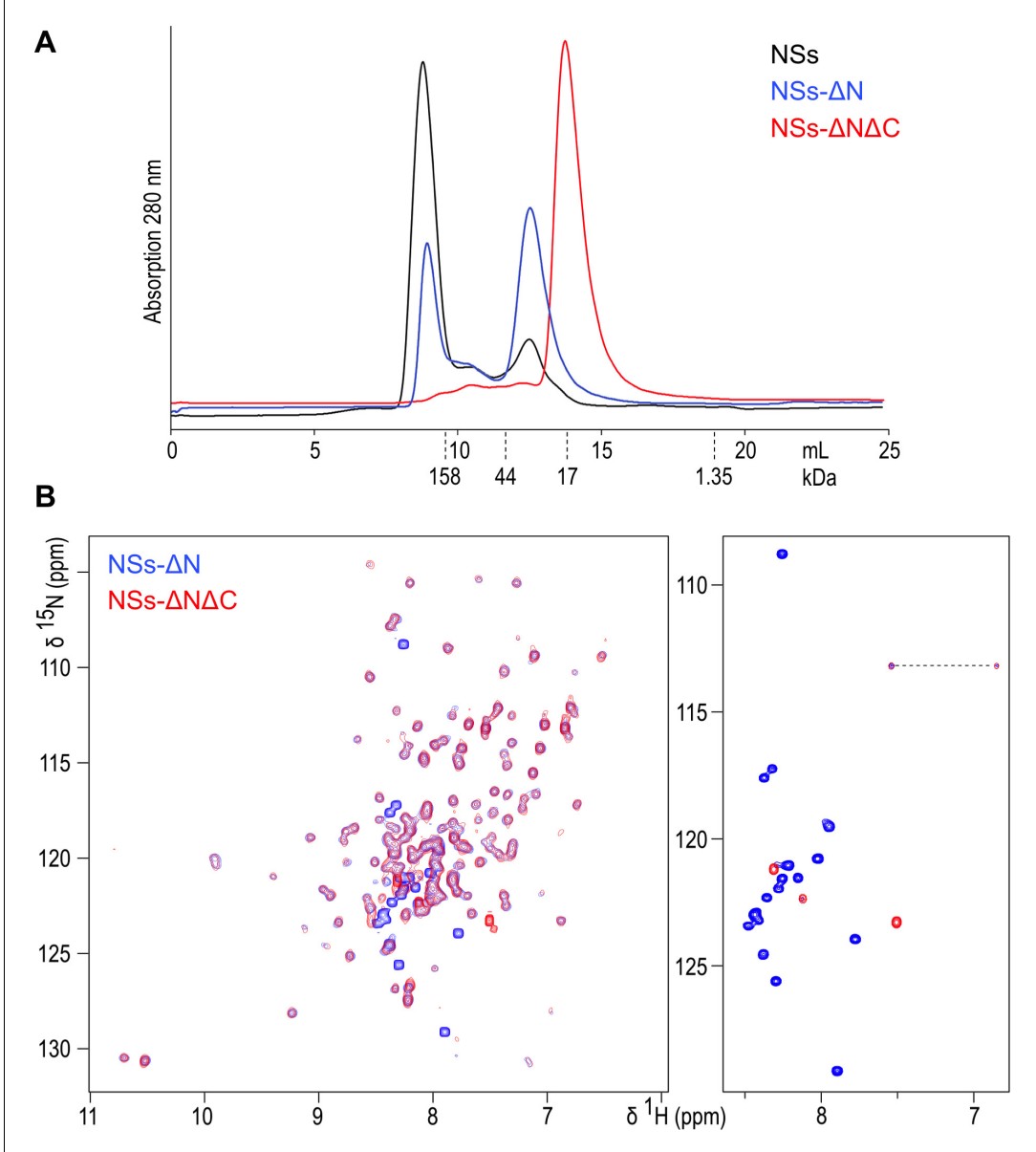

**Figure 1.** (A) Size exclusion chromatography elution traces of recombinantly expressed NSs variants, detected by UV absorption at 280 nm. Elution volumes for molecular mass markers used for calibration of the column are indicated with dashed lines. (B) $^1$H-$^{15}$N HSQC NMR spectra of NSs-ΔN (blue) and NSs-ΔNΔC (red) overlaid. The right-hand panel shows the central region of the spectra plotted at high contour levels, selecting a subset of intense crosspeaks representing highly dynamic amino acid residues, most of which are not present in NSs-ΔNΔC. The dashed line connects crosspeaks of one side-chain amide.

DOI: https://doi.org/10.7554/eLife.29236.003

The following figure supplement is available for figure 1:

**Figure supplement 1.** CD spectra of NSs.

DOI: https://doi.org/10.7554/eLife.29236.004

The structure was successfully phased from the anomalous sulfur signal in a single native crystal and refined to high quality (*Table 1*).

The NSs-ΔNΔC monomer, molecular mass 19,029 Da, measures approximately 30 × 30 × 50 Å. It is composed of eight α-helices with helices α1-α6 and α8 arranged around a long central helix α7. Two short $3_{10}$ helices are also present (*Figure 2*). Residues 234–244 are devoid of secondary structure and traverse approximately 30 Å of the NSs surface, packed against helices α3 and α7. Two

**Table 1.** Data collection and refinement statistics

**Data collection**

| | |
|---|---|
| Space group | $P6_422$ |
| Cell dimensions a, b, c (Å)<br>α, β, γ (°) | 123.8, 123.8, 174.0<br>90, 90, 120 |
| Beamline | Diamond i03 |
| Wavelength (Å) | 2.0 |
| Resolution (Å) | 107.83–2.19 (2.25–2.19) |
| Number of observed reflections | 1877448 |
| Number of unique reflections | 41684 |
| Completeness (%) | 99.9 (99.9) |
| $R_{merge}$ (%) * | 9.8 (223.4) |
| I/δ | 32.3 (1.6) |
| Multiplicity | 45.0 (18.2) |
| Anomalous completeness (%) | 99.9 (99.8) |
| Anomalous multiplicity | 23.7 (9.4) |
| Anomalous slope | 1.169 |
| **Refinement statistics** | |
| $R_{work}$ (%) † | 20.1 |
| $R_{free}$ (%) ‡ | 21.6 |
| Ramachandran favored (%) | 97.24 |
| Ramachandran allowed (%) | 99.08 |
| MolProbity score/percentile § | 1.61/97 |
| Averaged B-factor (Å$^2$) | 63.47 |
| Rmsd bond length (Å) | 0.905 |
| Rmsd bond angles (°) | 1.007 |

The values in parenthesis refer to the highest resolution shell.

*$R_{merge} = \frac{\sum_{hkl}\sum_i |I(hkl;i) - (hkl)>|}{\sum_{hkl}\sum_i (hkl;i)}$ where $I(hkl;i)$ is the intensity of an individual measurement of a reflection and $(hkl)>$ is the average intensity of that reflection.

†$R_{work} = \frac{\sum_{hkl} F_0 - F_c}{\sum_{hkl} F_0}$ where $F_0$ and $F_c$ are the observed and calculated structure factors, respectively.

‡ $R_{free}$ is $R_{work}$ with 5% of the observed reflections removed before refinement.

§ MolProbity score combines the clashscore, rotamer, and Ramachandran evaluations into a single score, normalized to be on the same scale as X-ray resolution. 100th percentile is the best among structures of comparable resolution; 0th percentile is the worst. For clashscore the comparative set of structures was selected in 2004, for MolProbity score in 2006 (**Chen et al., 2010**).

DOI: https://doi.org/10.7554/eLife.29236.005

NSs-ΔNΔC monomers constitute the asymmetric unit, with an interface formed by their α8 helices (**Figure 2—figure supplement 1**). There was no visible electron density for the first N-terminal and the five C-terminal amino acids; these residues are unaccounted for in the crystal structure.

Structural similarity searches with PDBeFold (**Krissinel and Henrick, 2004**) and DALI (RRID:SCR_013433) (**Holm and Rosenström, 2010**) did not identify significant structural homologues. We therefore conclude the NSs-ΔNΔC structure represents a novel fold.

## NSs-ΔNΔC in the crystal forms helical fibrils that are stabilized by extensive interfaces

In the crystal lattice NSs-ΔNΔC molecules were found to pack into a highly organized network of two fibril-like assemblies, referred to here as F1 and F2 (**Figure 3**), creating large solvent channels.

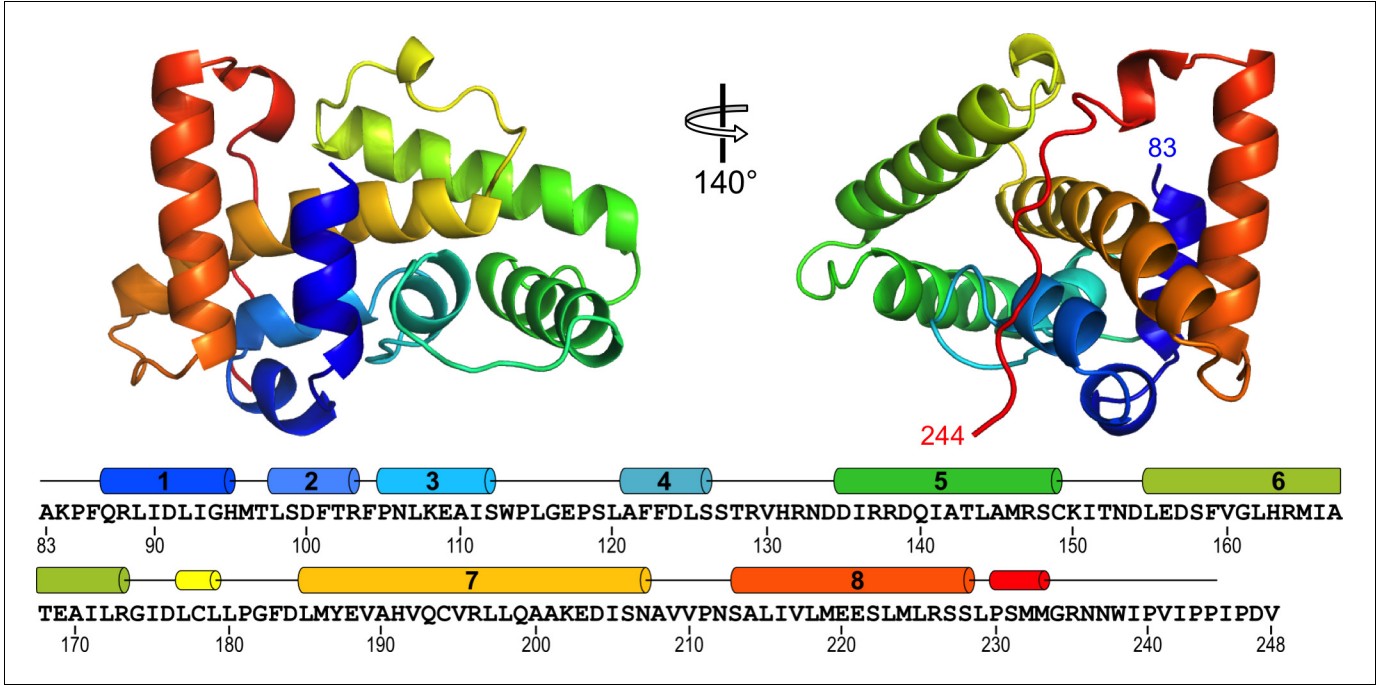

**Figure 2.** Structure of RVFV NSs-ΔNΔC. The helical arrangement of NSs shows a novel fold, comprised mainly of eight α-helices. The protein chain is colored in a rainbow spectrum from the N-terminus to the C-terminus. At the bottom, secondary structure elements are matched with the sequence of NSs-ΔNΔC (residues 83–248 of NSs). Numbered cylinders represent α-helices; narrower cylinders represent $3_{10}$ helices. Residues 244–248 are not defined in the crystal structure.

DOI: https://doi.org/10.7554/eLife.29236.006

The following figure supplement is available for figure 2:

**Figure supplement 1.** Asymmetric unit formed by two NSs molecules (colors as in *Figure 2*).

DOI: https://doi.org/10.7554/eLife.29236.007

The F1 and F2 crisscross pattern is produced by F1 fibrils running in parallel to each other and the longest unit cell edge, while F2 fibrils are oriented at an angle of 60° with respect to each other, perpendicular to F1 (*Figure 3—figure supplement 1*). F1 and F2 fibrils themselves are best described as being composed of repeating NSs tetramer units. Four different, partially overlapping (or alternative) tetramers T1, T2, T3 and T4 can be distinguished that differ in stability and interface areas, according to PISA interface analyses (*Krissinel and Henrick, 2007*), with T1 being the most stable tetramer (*Table 2*). F1 is an assembly of T1 tetramers stacked at 60° angles, while F2 is composed of alternating T1 and T2 tetramers arranged at 90° angles with respect to each other. The T1 interface that lies parallel to the F1 fibril is largely determined by hydrophobic packing of the α6 helices. The T1 interface perpendicular to the F1 fibril axis is defined by an extensive network of hydrogen bonds and salt bridges between the basic face of α1 (Lys84, Gln87, Arg88) and the acidic face of α8 (Glu220, Glu221, Ser228), leading to a large buried interface area of 4 × 852 Å² (*Figure 4*). The superficially very similar tetramer T2 in F2 fibrils is less stable. Here the molecules are rotated by about 8° with respect to the orientation of the equivalent monomers in T1, and form a twisted tetramer (*Figure 4—figure supplement 1*). The register of the α6-α6 interface is shifted, decreasing the buried surface area by 169 Å² compared to the same interface in T1. The total buried surface area is nearly 1000 Å² (17%) smaller in T2 than in T1. The B factors for T2 are also notably higher than for T1, suggesting that interfaces in T1, and therefore F1, possess a higher degree of conformational order, contributing to a higher stability of this assembly (*Figure 4—figure supplements 1,2*). B factors have been used to distinguish biologically relevant interfaces from crystal packing interfaces, which have been found to be characterized by relatively high B factors (*Liu et al., 2014*).

NSs-ΔNΔC T1 tetramers align at 60° angles in the F1 fibril, held by a network of hydrogen bonds clustered around the loop between helices α5-α6 (*Figure 4—figure supplement 1*). The 60° turn produces a left-handed double helix with twelve monomers per turn and a 34.8 nm pitch (*Figure 3*).

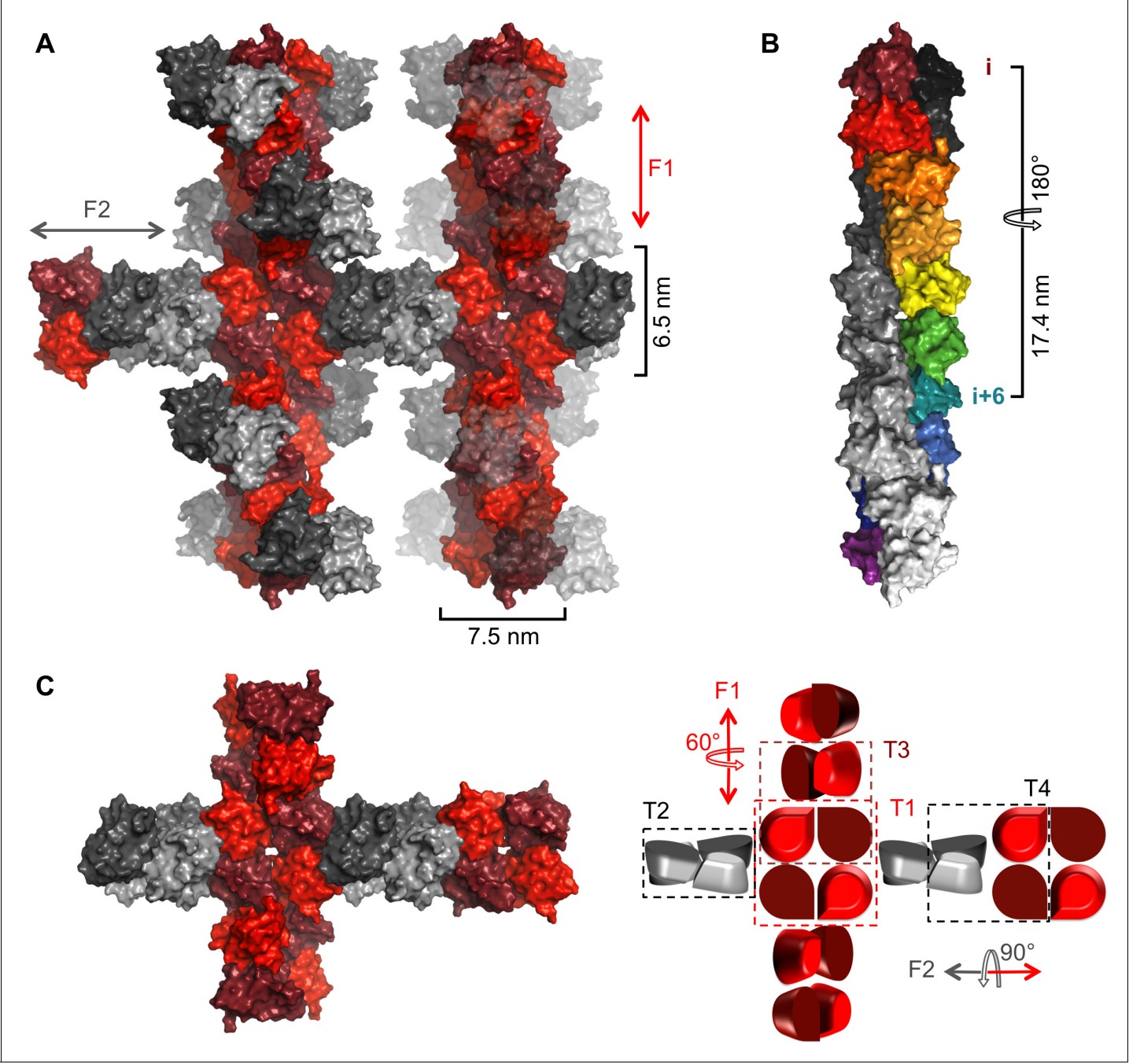

**Figure 3.** Crystal packing of NSs-ΔNΔC. (**A**) Two fibrillar arrangements, F1 and F2, result from assemblies of alternative tetramers. Two F1 fibrils (vertical) are shown with alternating shades of red for individual molecules. One F2 fibril (horizontal) is shown with alternating shades of gray for individual molecules in F2-specific tetramers (T2, T4). F2 fibrils emerge from each F1 tetramer (T1) but for clarity only two F2 (gray) NSs-ΔNΔC molecules are shown at each node. F2 units are shown in transparent mode on the right hand side F1 fibril to highlight the architecture of F1 fibrils. (**B**) The F1 fibril is a double helix of NSs-ΔNΔC monomers with 2-fold symmetry around the fibril axis and 6 molecules per 180° turn. A section of the F1 fibril containing five T1 tetramers is shown. One half of the double helix resulting from tetramer stacking is rainbow-colored, the other half is shown in shades of grey. (**C**) A simplified representation of a node between F1 and F2 fibrils highlights the architecture of F1 and F2 fibrils and the four different tetramers T1-T4.

DOI: https://doi.org/10.7554/eLife.29236.008

The following figure supplements are available for figure 3:

**Figure supplement 1.** Molecular packing in the NSs-ΔNΔC crystal lattice (wall-eye stereo image).

DOI: https://doi.org/10.7554/eLife.29236.009

*Figure 3 continued on next page*

*Figure 3 continued*

**Figure supplement 2.** Model of a full-length NSs fibril with the NSs-ΔNΔC F1 assembly as core (shown in surface representation).
DOI: https://doi.org/10.7554/eLife.29236.010
**Figure supplement 3.** Accessibility of the α8 helix (red) in F1 and F2 fibrillar crystal assemblies.
DOI: https://doi.org/10.7554/eLife.29236.011

## The architecture of NSs filaments in cells infected with RVFV

To gain insight into the architecture of nuclear NSs filaments, we infected Vero-E6 cells with the recombinantly produced parental RVFV strain MP12 as previously described (*Billecocq et al., 2008*; *Brennan et al., 2014*) and imaged cell nuclei containing filaments with transmission electron microscopy (TEM). Thick filaments of NSs, 0.5–1 μm in diameter, started forming about four hours post-infection and averaged to 1–2 filaments per nucleus. TEM reveals a substructure of these filaments (*Figure 5*). They appear to be composed of a bundle of tightly packed, extended parallel fibrils with diameters roughly ranging from 8 to 15 nm.

## Recombinant RVFV expressing NSs-ΔNΔC forms filaments in live cell nuclei

Recombinant RVFV encoding NSs-ΔNΔC (rMP12NSsΔNΔC) was rescued. Working stock of recombinant virus achieved similar titers ($1.15 \times 10^8$ pfu/mL) to parental MP12 ($2.00 \times 10^8$ pfu/mL) and no difference in plaque morphology was observed. Reverse transcription PCR analysis was performed to confirm the presence of the NSs truncation. Vero-E6 cells were infected with the recombinant rMP12NSsΔNΔC strain. Twenty-four hours after infection NSs-specific immunofluorescence staining showed prominent filaments in nuclei of infected cells (*Figure 6*). The morphology of these NSs-ΔNΔC filaments was indistinguishable from structures observed for the parental MP12 strain encoding full-length NSs. Fibrillation was observed only in about 30% of cells infected with rMP12NSsΔNΔC, compared to occurrence of filaments in nearly all cells infected with parental virus.

## Engineering of interfaces observed in NSs-ΔNΔC crystals prevents NSs filament formation in infected cells

To test if the fibrillar crystal packing observed for NSs-ΔNΔC represents a biologically relevant assembly underpinning intranuclear NSs filament formation, combinations of two key residues in three different crystal interfaces were mutated (*Figure 4*). Residues were chosen that were predicted to be critical for assembly of tetramers present in both or either F1 and F2 fibrils, but not essential for monomer stability. Three NSs variants were generated: NSs-ΔNΔC-muT1 (Arg88Asp, Ser228Ala), NSs-ΔNΔC-muT3 (Lys150Gly, Thr152Gly) and NSs-ΔNΔC-muT4 (Ile216Asp, Met219Ala). Disruption of the polar T1 interface, perpendicular to F1, would interrupt both T1 and T2, and therefore F1 and F2. Residue Arg88, forming two salt bridges to Glu220 and Glu221, was mutated to an aspartate. Ser228, forming a side-chain hydrogen bond with Gln87, was replaced by alanine. The T3 interface, unique to F1, was targeted through replacement of Lys150 and Thr152 with glycines. The hydrophobic T4 interface, formed by the α8 helices of the two molecules in the asymmetric unit of the crystals,

**Table 2.** Summary of PISA analyses of buried surface areas and ΔG values of the four unique NSs-ΔNΔC tetramers present in the crystal lattice. $\Delta G^{sol}$ and $\Delta G^{diss}$ are free energy of solvation (negative values indicate favorable process), and free energy of dissociation of assembly (positive values indicate stable association), respectively.

| | Buried surface [Å²] | $\Delta G^{sol}$ [kcal/mol] | $\Delta G^{diss}$ [kcal/mol] |
|---|---|---|---|
| T1 | 5631 | −33.1 | 10.6 |
| T2 | 4654 | −26.4 | 1.6 |
| T3 | 4358 | −25.2 | −3.0 |
| T4 | 4892 | −23.9 | 2.6 |

DOI: https://doi.org/10.7554/eLife.29236.015

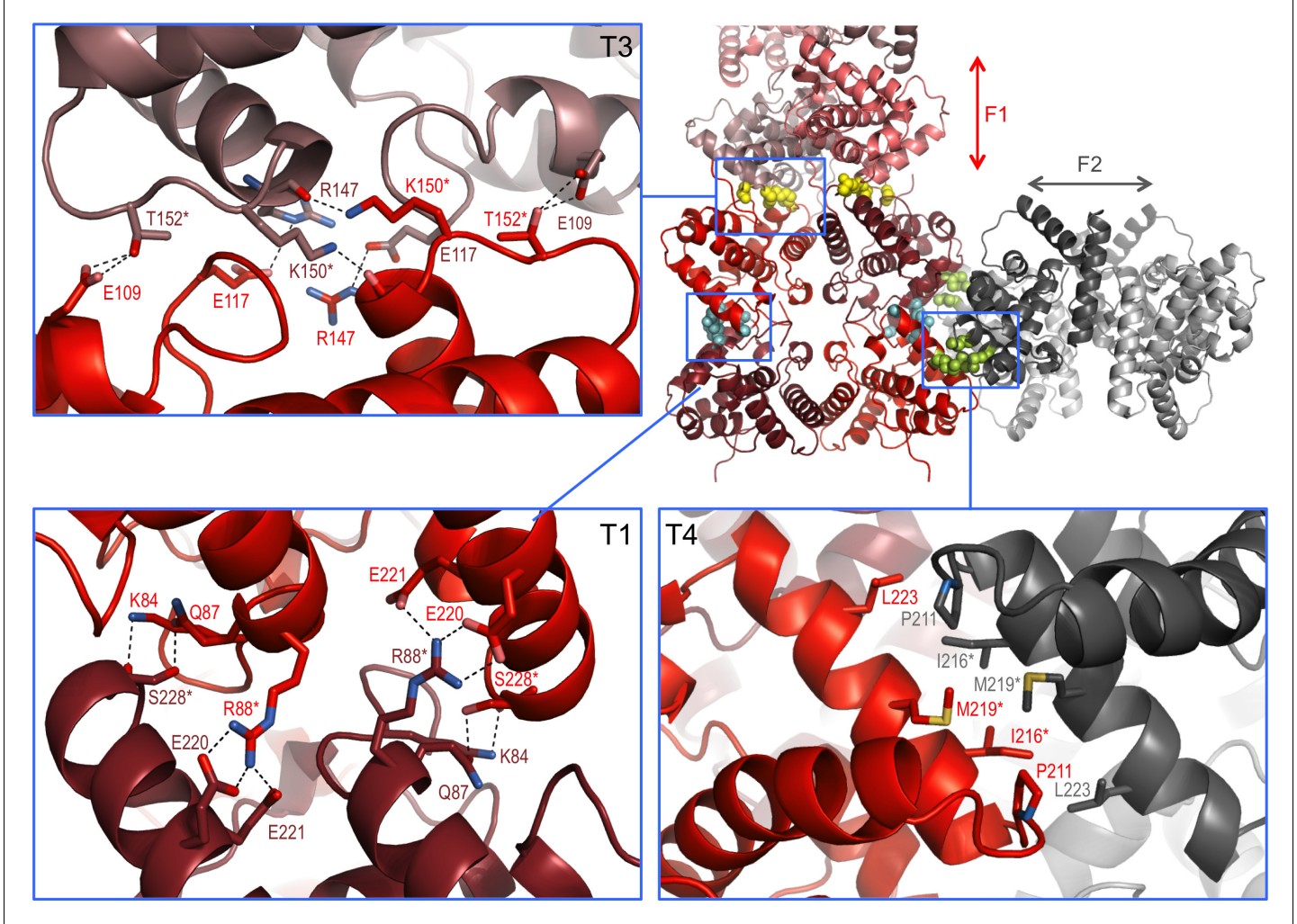

**Figure 4.** Interaction interfaces within the F1 and F2 fibrils observed in the NSs-ΔNΔC crystal. In the overview of an F1/F2 node (top right) key interface residues targeted by mutagenesis are shown as cyan, yellow and lime spheres for interfaces in T1, T3 and T4, respectively. Expanded regions highlight the residues involved in the three different interfaces. H-bond networks and salt bridges (dashed black lines) stabilize the interfaces for T3 and T1. The α8-α8 interface in T4 is defined by hydrophobic interactions. Residues changed by mutagenesis are marked by asterisks.

DOI: https://doi.org/10.7554/eLife.29236.012

The following figure supplements are available for figure 4:

**Figure supplement 1.** Comparison of T1 and T2 tetramers.

DOI: https://doi.org/10.7554/eLife.29236.013

**Figure supplement 2.** B-factor distribution in the NSs fibrillar crystal assembly.

DOI: https://doi.org/10.7554/eLife.29236.014

is unique to F2; it was changed by mutating Ile216 to aspartate and Met219 to alanine. These two residues contribute about 50% of the total buried surface area to the α8-α8 interface.

NMR spectroscopy was used to assess the effect of these mutations on the folded state of NSs-ΔNΔC in solution. [1]H spectra agreed with conservation of structure in all mutated proteins (*Figure 7—figure supplement 1*). All spectra showed the same signal dispersion and no indications of significant changes in structure, such as partial unfolding. High-field shifted methyl signals between −1 and 0 ppm were observed at the same chemical shifts in all spectra. Such signals belong to buried aliphatic side-chains, and conservation of their chemical shifts reflects preservation of the hydrophobic packing in the protein variants.

Three recombinant viruses encoding full-length NSs containing interface mutations were rescued, rMP12muT1NSs, rMP12muT3NSs, and rMP12muT4NSs. Recombinant viruses grew to similar titers

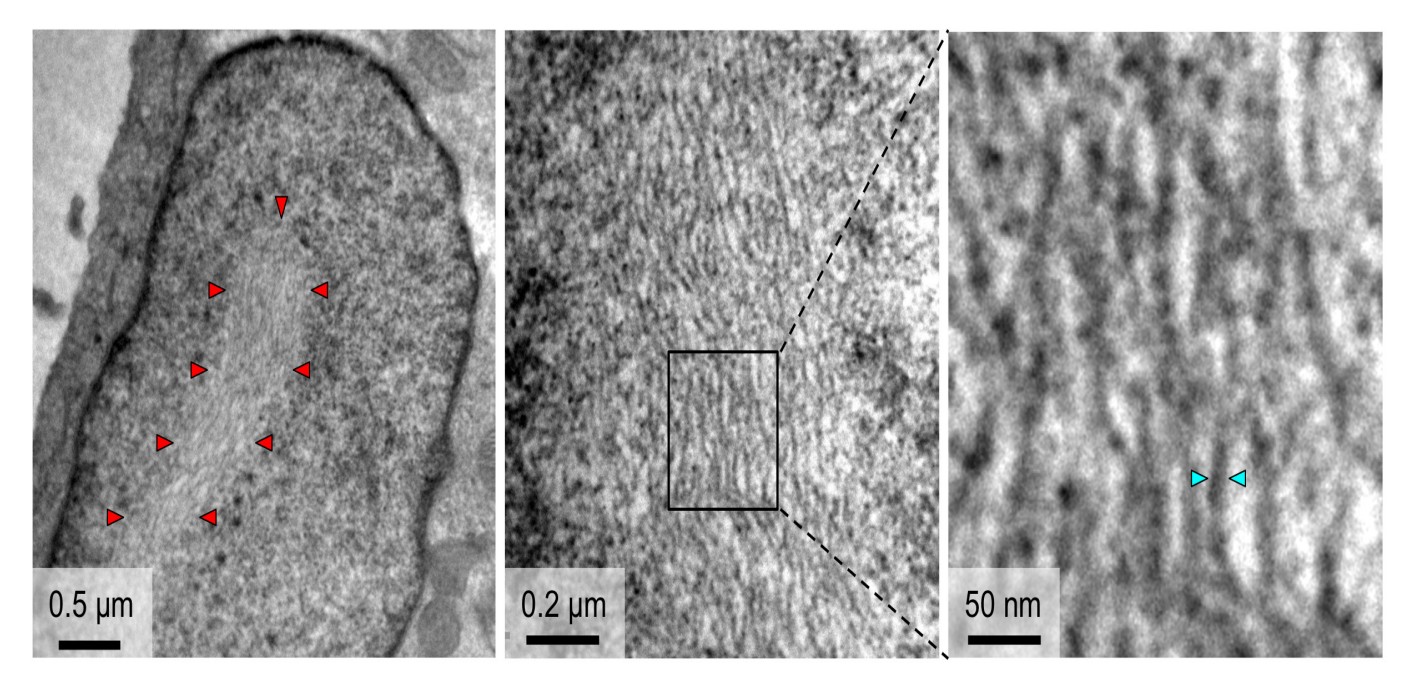

**Figure 5.** TEM observations of fibrils formed in the nuclei of cells infected with RVFV. The outline of an NSs filament within a cell nucleus is indicated in the left panel by red arrowheads. Higher magnifications (middle and right panels) reveal a substructure of NSs filaments, which are composed of bundles of parallel fibrils with estimated widths of 8–15 nm. Cyan arrowheads in the right panel indicate one fibril with a width of 10 nm.
DOI: https://doi.org/10.7554/eLife.29236.016

to that of the parental virus rMP12 and had indistinguishable plaque morphologies. Segment specific RT-PCR followed by Sanger sequencing was used to confirm the genomic composition of recombinant viruses. NSs-specific immunofluorescence staining showed the presence of filaments in nuclei of rMP12mut4NSs infected Vero-E6 cells (*Figure 7*). In contrast, no filaments were observed in nuclei of cells infected with rMP12muT1NSs and rMP12muT3NSs. In both cases NSs was detected as speckles or diffusely dispersed throughout the nuclei (*Figure 7*). These data indicate that all three NSs variants had retained the ability to be imported into nuclei. Mutating interfaces observed in the F1 crystal fibril clearly affected intranuclear NSs filament formation, whereas mutating an interface unique to the F2 fibril did not. Therefore, we conclude that the F1 crystal fibril is not only more stable than F2, but that it also represents the architecture of NSs in intranuclear filaments.

## Discussion

Innate immune antagonism is a common function of many viral non-structural proteins (*Randall and Goodbourn, 2008*), including NSs proteins from bunyaviruses (*Walter and Barr, 2011*). Non-structural viral proteins have been previously structurally characterized, such as the NS1 protein of influenza virus (*Bornholdt and Prasad, 2008*; *Liu et al., 1997*), VP35 of Ebola virus (*Leung et al., 2009*) or NS5A from hepatitis C virus (*Feuerstein et al., 2012*; *Tellinghuisen et al., 2005*). These studies have contributed to understanding both viral pathology and the innate immune system.

The NSs protein is the main virulence factor of RVFV and most bunyaviruses. RVFV lacking NSs cannot inhibit the interferon response and causes asymptomatic infection in mice (*Bouloy et al., 2001*; *Muller et al., 1995*; *Vialat et al., 2000*). Naturally occurring RVFV clone 13 encoding NSs lacking residues 16–198 (*Muller et al., 1995*), and a recombinant virus deficient of NSs, are currently candidates for live-attenuated vaccines for RVFV (*Bird et al., 2008*; *Bird et al., 2011*). The mode of action of RVFV NSs is understood to an extent. Several binding partners have been identified, and models for NSs function proposed (*Ly and Ikegami, 2016*). The underlying mechanisms of these

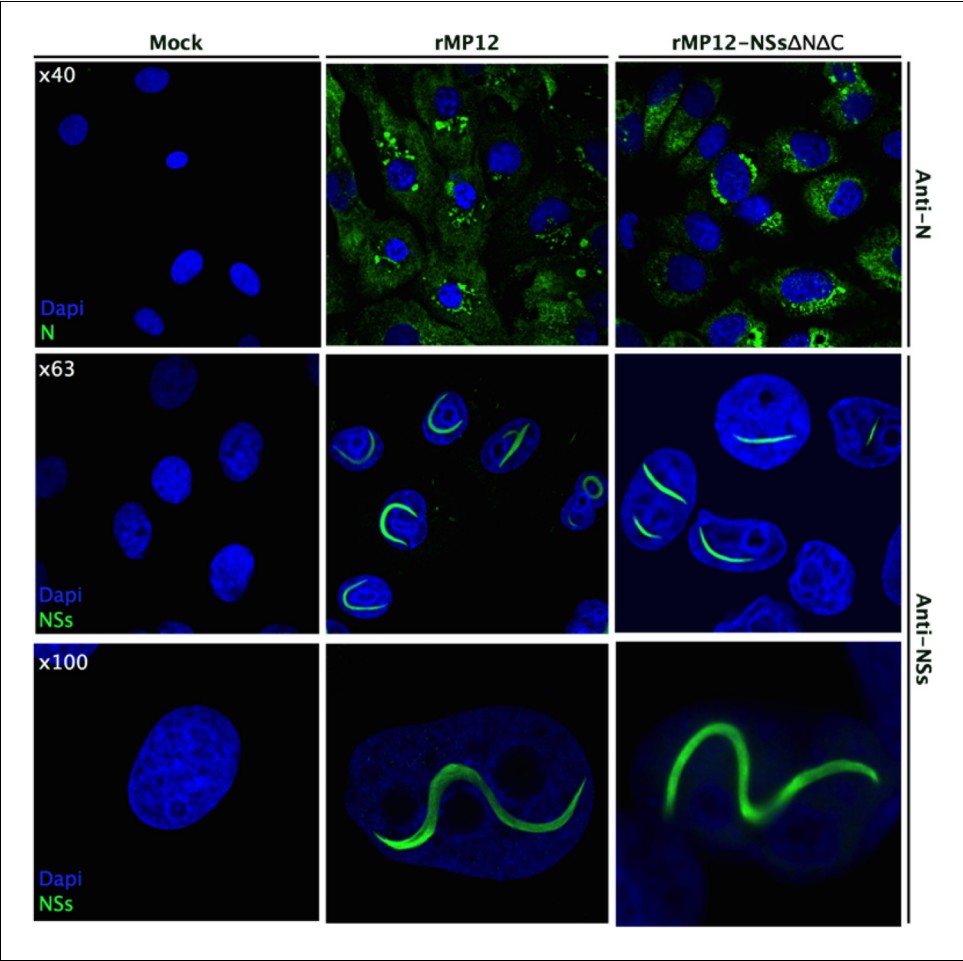

**Figure 6.** Intracellular localization of NSs in rMP12- or rMP12NSs-ΔNΔC-infected Vero-E6 cells. Cells were fixed 24 h p.i. and co-stained with either anti-N or anti-NSs antibodies (green) as indicated and DAPI (blue).
DOI: https://doi.org/10.7554/eLife.29236.017

interactions are however unknown, arguably because so far NSs proteins have largely evaded molecular and structural characterization. Here we report the first structure of a bunyaviral NSs protein.

Recombinant full-length RVFV NSs was found to form large aggregates in solution, and was not suitable for NMR or crystallography. Therefore, we designed a stable construct amenable to crystallization through double deletion of 82 N-terminal and 17 C-terminal residues (NSs-ΔNΔC). Both these regions had previously been linked to biological functions of NSs, particularly its intranuclear localization and fibrillation (*Yadani et al., 1999*; *Billecocq et al., 2004*; *Cyr et al., 2015*). However, we demonstrate that a recombinant virus containing NSs-ΔNΔC successfully induces filament assembly in nuclei of infected cells (*Figure 6*), indicating neither the truncated N- or C-terminal regions are required for fibrillation.

NSs does not contain any nuclear localization signals, thus nuclear import likely relies on host proteins such as the p44 subunit of the TFIIH transcription complex (*Le May et al., 2004*). Interaction sites for any such binding partners are predicted to be near the N-terminus, since mutation of a PxxP putative protein interaction motif (residues 29–32) results in retention of NSs in the cytoplasm (*Billecocq et al., 2004*). Our data, however, show that the N-terminal 82 residues of NSs (including the 29–32 motif) are dispensable for nuclear localization.

Deletion of 17 C-terminal residues was based on 2D NMR of NSs-ΔN showing the presence of an intrinsically unfolded region. Notably, this truncation matches one reported in a study where RVFV NSs lacking these C-terminal residues (encoded by recombinant Semliki Forest virus) was localized in nuclei of infected cells, but did not form filaments (*Yadani et al., 1999*). A more recent study

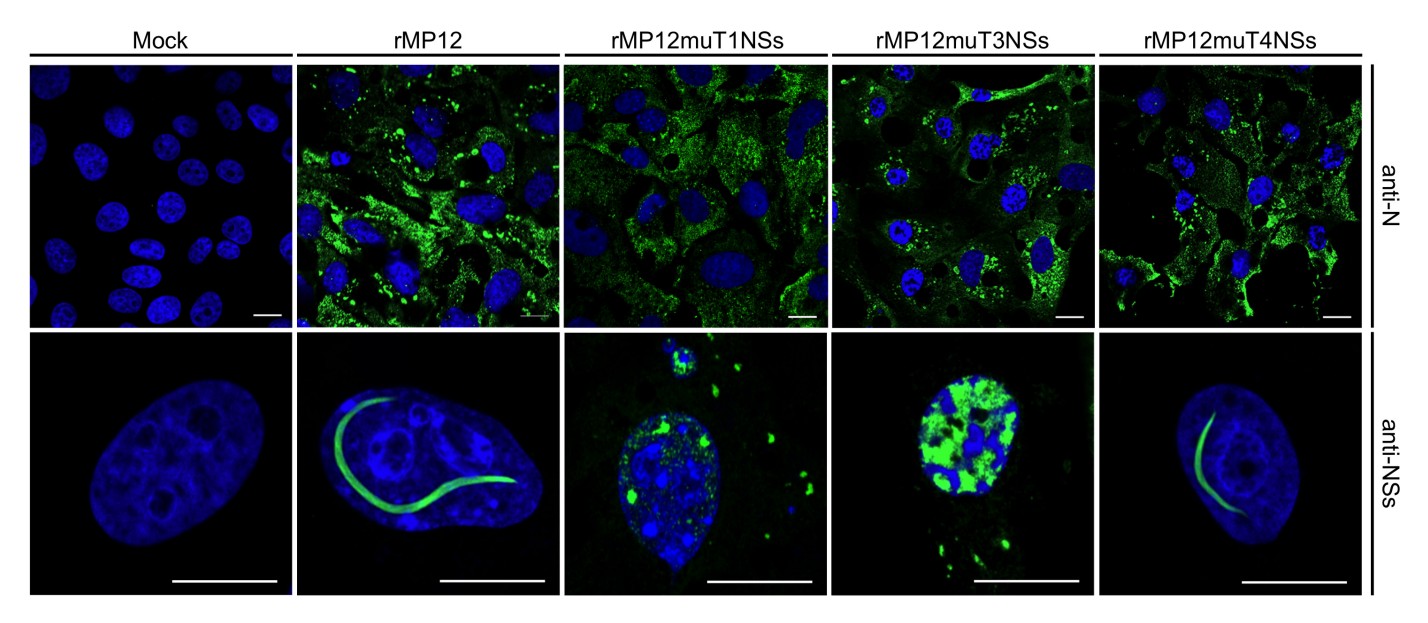

**Figure 7.** Intracellular localization of NSs in Vero-E6 cells infected with rMP12 or rMP12 variants encoding NSs variants with interface residue mutations (muT1, muT3, muT4). Cells were fixed 24 h p.i. and co-stained with either anti-N or anti-NSs antibodies (green) as indicated and DAPI (blue).
DOI: https://doi.org/10.7554/eLife.29236.018

The following figure supplement is available for figure 7:

**Figure supplement 1.** 1D 1 hr NMR spectra of NSs-ΔNΔC and interface mutation variants.
DOI: https://doi.org/10.7554/eLife.29236.019

suggests the 261-FVEV-264 motif is the binding site for the TFIIH subunit p62, and this interaction is required for filament formation (*Cyr et al., 2015*). While the C-terminal tail (including the FVEV motif) is missing in the NSs-ΔNΔC construct, we nevertheless observed intranuclear filaments indistinguishable from filaments in rMP12-infected cells. Therefore, neither the intrinsically disordered C-terminal nor the N-terminal regions of NSs are required for filament formation.

The discrepancies between published results and our findings regarding the terminal regions of NSs may be reconciled if a role for the N-terminal domain of NSs in self-association is considered. The N-terminal domain is included in the NSs constructs used in both studies cited above (*Cyr et al., 2015*; *Yadani et al., 1999*) that suggest a critical role of the C-terminal tail for NSs self-association. However, removal of the N-terminal domain had a clear effect on the oligomeric state of NSs-ΔN, where the C-terminal tail is present (*Figure 1A*). In crystal fibrils, NSs-ΔNΔC N- and C-termini are close to an apparent cleft in the middle of tetramer T1, which could accommodate the N-terminal domain, assuming full-length NSs fibrils contain the same core-domain arrangement. In which case, intermolecular interactions between monomer terminal regions are likely (*Figure 3—figure supplement 2*). Arguably, deletion of either N- or C-terminal regions results in non-native interactions that destabilize fibrils, artifacts abrogated in a doubly truncated construct.

Our data have implications for understanding the interaction between NSs and TFIIH. The model for NSs function previously proposed (*Cyr et al., 2015*) assumes initial binding of NSs to p62, with subsequent disintegration of the TFIIH complex, followed by sequestration of p44 into NSs filaments (*Le May et al., 2004*). This hypothesis is not supported by our findings, as NSs-ΔNΔC lacks the putative p62-binding motif and still yields filaments. The apparent similarity in size and shape of NSs-ΔNΔC filaments to full-length NSs does not allow conclusions about their composition. We cannot rule out the possibility that NSs-ΔNΔC filaments observed in vivo are lacking other proteins such as p44 possibly associated with full-length NSs filaments.

The NSs core domain structure reported here represents a novel fold. Comparison of the secondary structure distribution in NSs-ΔNΔC and secondary structure predictions for other phleboviral NSs proteins shows very good agreement, suggesting that, despite low to moderate sequence identity

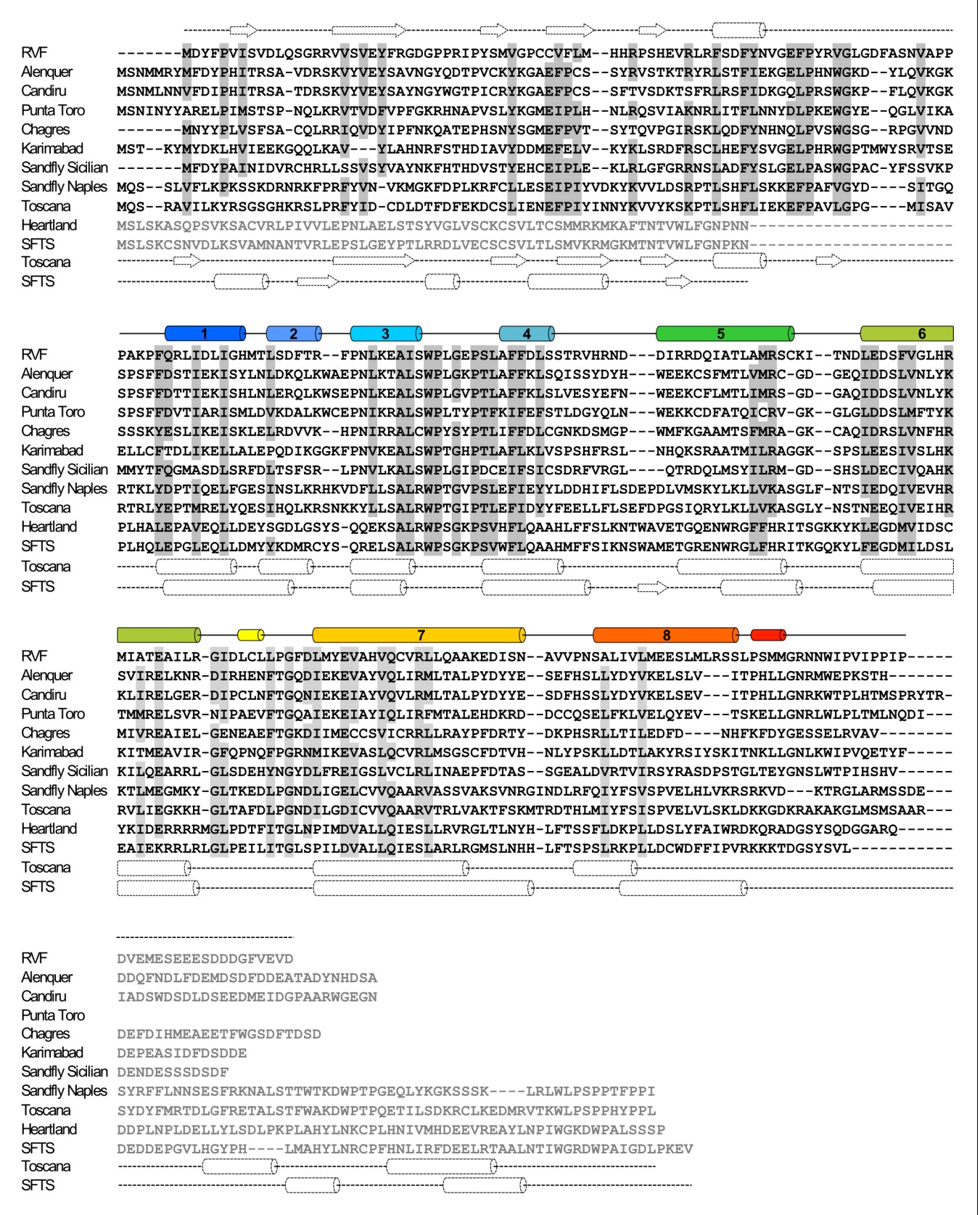

**Figure 8.** Sequence alignment of NSs proteins from human-pathogenic phleboviruses (SFTS, severe fever with thrombocytopenia syndrome virus). Secondary structure elements for RVFV-NSs are shown on top of the alignment panels. For the N-terminal domain (top panel) these are based on

*Figure 8 continued*

secondary structure prediction by JPred4 (*Drozdetskiy et al., 2015*) (dashed lines). Sequence similarity is indicated by gray shading for residues conserved or similar in at least 80% of sequences. Regions displayed in gray font are not aligned. Secondary structure predictions are also shown below each panel for Toscana virus and SFTS virus NSs.

DOI: https://doi.org/10.7554/eLife.29236.020

of this core region (15–32%), all phleboviral NSs proteins share a similar core domain fold (*Figure 8*). They do however vary greatly in their N- and C-terminal regions, which is particularly evident when comparing NSs from viruses transmitted by mosquitos and sandflies (e.g. RVFV, sandfly fever viruses) with viruses transmitted by ticks (e.g. Heartland phlebovirus and SFTS phlebovirus) (*Figure 8*).

The finding of highly organized fibrils in the NSs-ΔNΔC crystal lattice is intriguing due to the natural propensity of NSs for fibrillation in nuclei of RVFV-infected cells. Mutagenic analyses of interfaces defining the two types of crystal fibrils suggested that F1 likely represents the core architecture of NSs filaments, while F2 is a crystal packing artifact. T1 and T3 interface NSs variants were both detected in nuclei of infected cells. In immunofluorescence staining the muT3 interface variant appeared to be much more abundant than the muT1 variant, suggesting that the muT3 variant of NSs may be more stable than the muT1 interface mutant. This agrees with the predicted stability of the T1 tetramer (*Table 2*), making it less prone to degradation. To our knowledge, no other *Phlebovirus* causes intranuclear filament formation, and none of the residues stabilizing interfaces among NSs-ΔNΔC crystal fibrils is conserved in NSs sequences of phleboviruses other than RVFV.

Biological relevance of the F1 fibril is also supported by accessibility of the α8 helix in this assembly (*Figure 3—figure supplement 3*), which is partially buried in the F2-specific T1-T2 interface (*Figure 4*). This helix harbors a binding site for SAP30, a component of a repressor complex that regulates transcription of the interferon-β promoter (*Le May et al., 2008*). NSs arranged in the manner observed in NSs-ΔNΔC crystals would be able to bind to SAP30 and modulate SAP30 function.

PISA interface analysis indicates that potentially some NSs-ΔNΔC oligomers, but neither F1 nor F2 fibrils would be stable in solution. This agrees with the observation that NSs is distributed throughout the cytoplasm at early time points in cell infection experiments, and only forms distinct filaments inside nuclei. Fibrillation therefore depends on factors present in the nucleus, but not the cytosol. Essential factors could be binding partners, but also other conditions that shift the association equilibrium of NSs towards fibrils such as the excluded volume effect, or molecular crowding, which plays a role in compartmentalization and architecture of cell nuclei (*Hancock, 2004*; *Richter et al., 2008*). It is tempting to speculate that intranuclear crowding conditions are mimicked in crystallization where an excluded volume effect is induced by the precipitant polyethylene glycol (PEG).

RVFV is one of the most dangerous human pathogens with the potential to cause wide epidemics in the near future. The first high-resolution structure for the main RVFV virulence factor NSs and other data presented here will facilitate a deeper understanding of NSs function. Ongoing research will help determine if NSs filament assembly is required for virulence, a question that this study does not attempt to answer. NSs proteins of phleboviruses and other members of the *Bunyavirales* order are highly diverse in sequence, but all play roles in suppressing the innate immune response through various mechanisms (*Ly and Ikegami, 2016*). In infected cells, NSs proteins are located in the nuclei and cytosol, or the cytosol only. Currently no other NSs protein is known to cause filaments characteristic for RVFV. To establish whether this unique property of RVFV NSs is related to virulence may require animal models, since NSs is dispensable for viral replication and in vitro cell infection, but is required to cause pathology (*Bouloy et al., 2001*). The structure-based insights presented here should facilitate such studies. Development of modulators of fibril formation may be an attractive option for new therapeutics, especially given the conservation of interface residues in RVFV isolates.

## Materials and methods

### Protein expression and purification

Expression constructs were prepared by cloning RVFV strain MP12 cDNA comprising full-length (residues 1–265), ΔN (residues 83–265), and ΔNΔC (residues 83–248) regions of NSs into a modified

pMal-C2x vector (New England Biolabs, MA, USA) containing an N-terminal hexahistidine tag downstream of the *malE* (MBP) gene and a tobacco etch virus (TEV) protease sequence. Proteins were expressed using *Escherichia coli* BL21 (DE3). Cells were grown at 37°C in Luria-Bertani broth to an $OD_{600}$ of 0.6. Protein expression was induced with a final concentration of 0.5 mM isopropyl β-d-1-thiogalactopyranoside and allowed to proceed overnight at 18°C. Cells were pelleted by centrifugation and lysed in the presence of 10 mM phosphate buffer, 300 mM NaCl, 20 mM imidazole, pH 7.2 using sonication. Soluble proteins were separated from cell debris by centrifugation. Soluble protein was purified by nickel metal-affinity chromatography (IMAC). After binding, resin was washed extensively with 10 mM phosphate buffer, 2 M NaCl, 20 mM imidazole, pH 7.2 in order to dissociate any bound DNA. MBP-NSs was eluted in 10 mM phosphate buffer, 300 mM NaCl, 250 mM imidazole, pH 7.2, and dialyzed overnight at 4°C in 10 mM phosphate buffer, 300 mM NaCl, 2 mM DTT, pH 7.2 with simultaneous cleavage by TEV protease. MBP was separated from NSs by reverse nickel IMAC. NSs was further purified by size exclusion chromatography on a 16/60 S75 Superdex column (GE Healthcare, UK) in 10 mM phosphate, 300 mM NaCl, 1 mM DTT, pH 7.2.

Interface mutation variants of NSs-ΔNΔC were generated through site-directed mutagenesis. NSs-ΔNΔC-muT1 (Arg88Asp, Ser228Ala) was generated in two stages of amplification, while NSs-ΔNΔC-muT3 (Lys150Gly, Thr152Gly) and NSs-ΔNΔC-muT4 (Ile216Asp, Met219Ala) were made in single steps. Interface mutation variants were expressed and purified as previously described for wild-type NSs-ΔNΔC.

## Crystallization and structure determination

NSs 83–248 (NSs-ΔNΔC) was dialyzed into 20 mM HEPES, 150 mM NaCl, 1 mM DTT, pH 7.2, concentrated to 30 mg mL$^{-1}$. Protein crystals were grown at 4°C by sitting-drop vapor diffusion in 0.1 M Tris-HCl pH 8.5, 8% (w/v) PEG 10,000 with protein and reservoir combined in a 2:1 ratio. Crystals of NSs-ΔNΔC grew as elongated hexagons and reached their maximum size of about $0.3 \times 0.1 \times 0.1$ mm after two weeks. Crystals were cryo-protected by brief immersion in reservoir supplemented with 30% (v/v) glycerol then flash frozen in liquid nitrogen. Native and anomalous data were collected using a 2.0 Å wavelength beam at the Diamond Light Source (I03 beamline). Diffraction over a 500° angle yielded a high-multiplicity dataset with maximum resolution of 2.2 Å. Crystals indexed in space group $P6_422$ with cell dimensions a = 123.8 Å, b = 123.8 Å, c = 174 Å; α = 90°, β = 90°, γ = 120°. Data were indexed and integrated using XDS, scaled using XSCALE (*Kabsch, 2010*), and merged with Scala (*Evans, 2006*). The structure was solved by single anomalous dispersion (SAD); phase determination from native sulfur anomalous signal was performed using SHELX (*Sheldrick, 2010*). PHENIX (RRID:SCR_014224) AutoBuild was used for initial automated model-building (*Terwilliger et al., 2008*). Density modification and model refinement was performed with REFMAC5 (RRID:SCR_014225) (*Murshudov et al., 1997*) and Coot (RRID:SCR_014222) (*Emsley et al., 2010*). Structure validation was conducted with the MolProbity (RRID:SCR_014226) server (*Chen et al., 2010*). Data collection and refinement statistics are listed in *Table 1*.

## NMR spectroscopy

NSs constructs were isotopically labeled by expression in M9 minimal media, supplemented with 1 g L$^{-1}$ of $^{15}NH_4Cl$ and purified as described above for native proteins. NMR samples typically contained 0.3 mM protein in 10 mM phosphate, 50 mM NaCl, 0.25 mM DTT, pH 7.2, 5% (v/v) $D_2O$. $^1$H-$^{15}$N HSQC spectra were recorded on a Bruker DRX500 spectrometer equipped with a 5 mm TXIz probe at 15°C. Spectra were processed with NMRPipe (*Delaglio et al., 1995*) and analyzed with CCPN Analysis 2 (*Vranken et al., 2005*). For interface mutation variants of NSs-ΔNΔC, samples contained 0.03–0.25 mM protein in 10 mM phosphate, 150 mM NaF, 1 mM DTT, pH 7.2, 5% (v/v) $D_2O$. One-dimensional $^1$H spectra were recorded with a spectral resolution of 1.7 Hz on a Bruker Ascend 700 MHz spectrometer equipped with a Prodigy TCI probe at 22°C. The spectra were processed and analyzed using Bruker Topspin 3 (RRID:SCR_014227).

## Cells and viruses

Vero-E6 cells (RRID:CVCL_0574) were purchased from the European Collection of Authenticated Cell Cultures (ECACC), and tested negative for mycoplasma. Cells were grown in Dulbecco's modified Eagle's medium (DMEM) supplemented with 10% (v/v) fetal calf serum. Cell lines were grown at

37°C with 5% $CO_2$ unless otherwise stated. Recombinant viruses containing truncated or mutated NSs proteins were generated using reverse genetics as previously described (*Brennan et al., 2014*; *Le May et al., 2008*). Stocks of recombinant viruses were grown in BHK-21 cells at 33°C by infecting at a multiplicity of infection (MOI) of 0.01 and harvesting the culture medium at 5–7 days post infection (p.i.) All experiments with infectious virus were conducted under containment level 3 conditions. RT-PCR analysis of p1 stock viruses was performed to confirm the correct configuration of the recombinant virus S RNA.

## Virus titration by plaque assay

BHK-21 (clone 13) cells (RRID:CVCL_1915) were purchased from the European Collection of Authenticated Cell Cultures (ECACC), and tested negative for mycoplasma. Cells were infected with serial dilutions of virus and incubated under an overlay consisting of Glasgow minimum essential medium (GMEM) supplemented with 2% (v/v) newborn calf serum and 0.6% Avicel (w/v) (FMC BioPolymer, Philadelphia, PA) at 37°C for 4 days. Cell monolayers were fixed with 4% (w/v) formaldehyde and plaques were visualized by Giemsa staining.

## Immunofluorescence microscopy of infected cells

Vero-E6 cells were grown on glass coverslips (13 mm diameter), infected with recombinant or parental RVFV at an MOI of 5 and fixed at 24 h p.i. in 4% (w/v) formaldehyde in PBS. After permeabilization with 0.1% (v/v) Triton X-100 in PBS, cells were stained with specific primary antibodies, followed by secondary antibody conjugates. Localization of the fluorescently labeled proteins was examined using a Zeiss LSM-710 confocal microscope. Images in *Figures 6* and *7* are representatives of two independent infection experiments, with two repeats for each antibody.

## Thin-section electron microscopy

Monolayer cultures of Vero-E6 cells were seeded in 6-well plates and infected with RVFV MP12 for 24 h p.i. at an MOI of 3, and subsequently fixed with 2.5% (v/v) glutaraldehyde overnight at 4°C. Cells were scraped and pelleted by centrifugation followed by fixation with 1% (w/v) osmium tetroxide (TAAB Labs, UK) and staining with 2% (w/v) aqueous uranyl acetate for 1 hr at room temperature. Cells were then harvested into PBS and pelleted through 1% (w/v) SeaPlaque agarose (Sigma, UK) at 45°C. The agar was set at 4°C and cell pellets were cut into ~1 mm cubes, which were dehydrated through a graded alcohol series (30–100% (v/v)) and embedded in Epon 812 resin (TAAB Labs, UK) followed by polymerization for 3 days at 65°C. Thin sections of 120 nm were cut with a UC6 ultramicrotome (Leica Microsystems, Germany) and examined with a JEOL 1200 EX II electron microscope and images were recorded on a Gatan Orius CCD camera.

## Accession code

The NSs-ΔNΔC protein structure, and the data used to derive these, have been deposited at PDBe (RRID:SCR_004312) with accession number 5OOO.

## Acknowledgements

We acknowledge the Diamond Light Source for access to Data Collection facilities.

## Additional information

### Funding

| Funder | Grant reference number | Author |
|---|---|---|
| Medical Research Council | MDTG11 - MR/J500343/1 | Michal Barski Ulrich Schwarz-Linek |
| Wellcome | 099220/Z/12/Z | Benjamin Brennan Richard M Elliott |
| Royal Society of Edinburgh | Caledonian Scholarship | Ona K Miller |

The funders had no role in study design, data collection and interpretation, or the decision to submit the work for publication.

## Author contributions

Michal Barski, Conceptualization, Investigation, Visualization, Writing—original draft, Writing—review and editing; Benjamin Brennan, Resources, Investigation, Visualization, Writing—review and editing; Ona K Miller, Investigation, Writing—review and editing; Jane A Potter, Formal analysis, Investigation; Swetha Vijayakrishnan, Investigation, Visualization, Writing—review and editing; David Bhella, Resources, Supervision; James H Naismith, Formal analysis, Methodology; Richard M Elliott, Conceptualization, Supervision, Funding acquisition; Ulrich Schwarz-Linek, Conceptualization, Supervision, Funding acquisition, Investigation, Writing—original draft, Project administration, Writing—review and editing

## Author ORCIDs

Michal Barski (iD) http://orcid.org/0000-0002-4663-6915
Benjamin Brennan (iD) http://orcid.org/0000-0003-4707-726X
Ona K Miller (iD) https://orcid.org/0000-0003-2486-7680
Ulrich Schwarz-Linek (iD) http://orcid.org/0000-0003-0526-223X

## Decision letter and Author response

Decision letter https://doi.org/10.7554/eLife.29236.024
Author response https://doi.org/10.7554/eLife.29236.025

## Additional files

### Supplementary files

• Transparent reporting form
DOI: https://doi.org/10.7554/eLife.29236.021

### Major datasets

The following dataset was generated:

| Author(s) | Year | Dataset title | Dataset URL | Database, license, and accessibility information |
|---|---|---|---|---|
| Barski M, Potter J, Schwarz-Linek U | 2017 | Structure of the Rift Valley fever virus NSs protein core domain | http://www.rcsb.org/pdb/explore/explore.do?structureId=5OOO | Publicly available at the RCSB Protein Data Bank (accession no:5OOO) |

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
