## [Decision Letter]

Thank you for submitting your article "Rift Valley fever virus NSs protein core domain structure suggests molecular basis for pathogenic nuclear filaments" for consideration by *eLife*. Your article has been reviewed by three peer reviewers, and the evaluation has been overseen by John Kuriyan as the Senior Editor and Reviewing Editor. The following individuals involved in review of your submission have agreed to reveal their identity: Yorgo Modis (Reviewer #1); Jason Mclellan (Reviewer #2), Richard J Kuhn (Reviewer #3).

In this study Barski et al. describe the crystal structure of the non-structural protein NSs from Rift Valley fever virus (RVFV) of the Bunyaviridae. NSs is an important virulence factor that impedes interferon production by at least three mechanisms: binding to the p44 subunit of TFIIH; binding to SAP30; and targeting the RNA-dependent protein kinase PKR for degradation. Moreover, RVFV NSs forms nuclear filaments, which have been reported to associate with pericentromeric DNA, potentially causing chromosomal segregation defects.

The authors identified a soluble monomeric fragment of NSs using gel filtration and NMR. The 2.2 Å structure of the NSs core was determined by phasing with sulfur anomalous signals, and was found to be a novel fold consisting of 8 α-helices. In the crystal, NSs forms two types of fibrils, each with two-start helical symmetry. The authors demonstrated that these fibrils have diameters similar to those observed by thin-section EM in Vero-E6 cells infected with RVFV. Immunofluorescence microscopy showed that RVFV containing the crystallized fragment of NSs formed similar nuclear filaments as wild-type RVFV. However, viruses containing an NSs gene with either of two structure-based mutations aimed at disrupting the "F1"-type fibril-forming contacts did not form any nuclear filaments in infected cells. The viruses bearing the mutant proteins grew to similar titers and had similar plaque morphologies as the wild-type. The authors conclude that the F1 fibril-forming interface in the crystal (the stronger of the two interfaces) corresponds to the physiological interface responsible for NSs filament formation during RVFV infection. The authors note that the N- and C-terminal tails containing putative protein interaction regions, including the putative SAP30 interaction region, are exposed on the surface of F1 NSs fibrils.

This manuscript is clear and very well written. The structural studies are excellent, as is the analysis of the fibrils found in the lattice. The figures are clear, and they helped to convey the three-dimensional lattice and fibril interactions. The structure of the RVFV bunyaviral NSs protein is important and will help the field better understand the function of this key virulence factor. The identification of a soluble and crystallizable NSs fragment that can form physiologically relevant fibril contacts in the crystal is a technical achievement.

The reviewers recognize that a weakness of this study is that it does not address the question of whether filament formation is important for NSs function. The authors speculate that the NSs fibrils may promote infection by sequestering NSs binding partners, but it remains unclear whether NSs filaments have any proviral activity beyond that of soluble NSs. Indeed, the finding that the NSs-mutant RVFVs do not form NSs filaments but have the same infectivity as wild-type virus in cell culture, and the lack of conservation of fibril-forming residues in NSs in other phleboviruses (despite predicted conservation of the overall fold of NSs in the genus), suggest that NSs filament formation may not play a fundamental role in the viral replication cycle. Additionally, although the site-directed mutagenesis lends support to the authors' conclusion that the F1 fibrils represent the architecture of NSs filaments in vivo, the data do not rule out the possibility that the native full-length NSs filaments have an alternative conformation in vivo. While recognizing these limitations of the present work, we have concluded that the paper represents a sufficient level of new information concerning this important virus for publication in *eLife*.

Major Comment:

Given the reservations expressed above, we ask that the title be modified to remove the adjective "pathogenic" with regard to the nuclear filaments. In addition, we also ask that the Discussion section of the paper be edited to bring up the limitations of the study, as discussed above, making it clear to the reader that the relevance of the fibrils for diseases causation remains to be established. We note that the structural information provided here should facilitate such studies, a point that the authors could make to balance the principal weakness of the work, while not ignoring that weakness.

Other comments to address:

1) What is the rationale for naming the tetramers T1-T4? Since T1 and T3 are discussed at length and have a broadly similar more planar arrangement, it would seem preferable to rename T3 to T2, use T3 and T4 for the non-planar tetramers that are discussed less (currently T2 and T4).

2) Subsection “NSs-ΔNΔC in the crystal forms helical fibrils that are stabilized by extensive interfaces”, end of first paragraph. High B-factors do not necessarily reflect conformational instability of a crystallized component.

3) Discussion, ninth paragraph. This section of the Discussion is simply restating the results and should be deleted or shortened significantly.

4) Discussion, sixth paragraph: change "almost inevitable" to "likely" (or similar).

5) Figure 3. The panel on the right is very effective.

6) Figure 6. In certain views (e.g. certain bottom panel) the NSs filaments have the appearance of a flattened ribbon or sheet rather than a string or worm-shaped filament. The details of the filament assembly may be worth looking at in greater detail.

7) Figure 8. This figure does not convey any information that cannot be conveyed in words and the modeled C-terminal tails could be confusing to some readers. The figure could be deleted (or moved to the supplement).

8) Figure 3—figure supplement 1. The figure would be more effective if the view were rotated to move the axis of the gray rods closer to perpendicular to the paper. Also, the more widely used wall-eye stereo view should be used instead of cross-eye.

9) Introduction, last paragraph: should "stabilize" be "destabilize"?

10) Subsection “Crystal structure shows the NSs core domain adopts a novel fold”, first paragraph: include "resolution" or "Bragg spacing" after 2.2 Å.

11) Subsection “Crystal structure shows the NSs core domain adopts a novel fold”, first paragraph, Table 1, and elsewhere: the "P" in P6422 should be italicized.

12) Figure 2 legend: It is stated that the protein chain is colored in a blue-to-red spectrum, but it would be more accurate to state that it is colored as a rainbow. A blue-to-red spectrum could just be blue-purple-red.

13) Introduction, first paragraph: "requiring urgent attention" was already mentioned in the same sentence and should be deleted.

14) – Introduction, fourth paragraph: mention that this protein is an ORFan therefore having limited sequence similarity. Do the other Phleboviruses use these proteins and how related are they within the Bunya's?

15) Subsection “Crystal structure shows the NSs core domain adopts a novel fold”, second paragraph: – – it would be nice to mention the molecular mass of the protein somewhere.

16) Figure 2. – Include the numbers and the N and C termini distinctions in the picture of the structure.

---

## [Author Response]

Major Comment:Given the reservations expressed above, we ask that the title be modified to remove the adjective "pathogenic" with regard to the nuclear filaments. In addition, we also ask that the Discussion section of the paper be edited to bring up the limitations of the study, as discussed above, making it clear to the reader that the relevance of the fibrils for diseases causation remains to be established. We note that the structural information provided here should facilitate such studies, a point that the authors could make to balance the principal weakness of the work, while not ignoring that weakness.

The title was changed as suggested. We added a statement to the end of the Discussion to highlight that this study did not attempt to establish a role of NSs filament formation for pathogenicity, and that this is a clear limitation of our work. Establishing the role of filament formation for virulence, which may require animal models, is beyond the scope of our structural and cell-based approach. For instance the observation of similar plaque sizes caused by virus variants studied here does not shed light on the role of NSs filaments in pathogenesis, since NSs protein is not required for viral replication and cell infection. However lack of NSs results in loss of virulence in animal models. We believe the structural data and insight into the molecular determinants of NSs polymerization will greatly facilitate future studies aimed at understanding the role of NSs filaments. It will help answering the questions why RVFV appears to be the only virus inducing such filaments, and if this is in any way linked with the unique spectrum of pathologies associated with RVFV infections, in particular fetal deformities and abortions, which are a hallmark of RVFV.

Other comments to address:1) What is the rationale for naming the tetramers T1-T4? Since T1 and T3 are discussed at length and have a broadly similar more planar arrangement, it would seem preferable to rename T3 to T2, use T3 and T4 for the non-planar tetramers that are discussed less (currently T2 and T4).

There was no particular rationale behind numbering the tetramers. As suggested the tetramers have been renamed in the text and figures. Also NSs protein and virus variants previously named “muT2” have been renamed “muT3” to reflect mutations affecting the interface of the (now) T3 tetramer.

2) Subsection “NSs-ΔNΔC in the crystal forms helical fibrils that are stabilized by extensive interfaces”, end of first paragraph. High B-factors do not necessarily reflect conformational instability of a crystallized component.

We agree, however B factors in a crystal assembly do reflect the quality of packing, or the definition of interfaces. A lower degree of order within an interface would make this interface less stable. We have changed the text to avoid any ambiguity, and added a reference to highlight that B factors have been used to distinguish biological interfaces from packing artifacts.

3) Discussion, ninth paragraph. This section of the Discussion is simply restating the results and should be deleted or shortened significantly.

This section was shortened significantly.

4) Discussion, sixth paragraph: change "almost inevitable" to "likely" (or similar).

Changed as suggested.

5) Figure 3. The panel on the right is very effective.

Thank you.

6) Figure 6. In certain views (e.g. certain bottom panel) the NSs filaments have the appearance of a flattened ribbon or sheet rather than a string or worm-shaped filament. The details of the filament assembly may be worth looking at in greater detail.

The appearance of filaments in the immunostaining images is most likely an artifact of microscopy of large structures occupying many focal planes; it depends on the choice of focal plane. Data presented here do not allow conclusions about the three-dimensional shape of the filaments, which we agree deserves attention in the future.

7) Figure 8. This figure does not convey any information that cannot be conveyed in words and the modeled C-terminal tails could be confusing to some readers. The figure could be deleted (or moved to the supplement).

We believe the figure is not essential for the paper, but useful for highlighting how a full-length NSs fibril might be based on the NSs-ΔNΔC core fibril. The C-terminal tails have been shown to be natively unfolded, and therefore adding them to the fibril model as extended randomly oriented extensions is justifiable. The figure is now a supplementary figure (Figure 3—figure supplement 2).

8) Figure 3—figure supplement 1. The figure would be more effective if the view were rotated to move the axis of the gray rods closer to perpendicular to the paper. Also, the more widely used wall-eye stereo view should be used instead of cross-eye.

We improved the figure so the gray rods align better with the fibrils, and changed to wall-eye stereo.

9) Introduction, last paragraph: should "stabilize" be "destabilize"?

Changed to make this statement unambiguous.

10) Subsection “Crystal structure shows the NSs core domain adopts a novel fold”, first paragraph: include "resolution" or "Bragg spacing" after 2.2 Å.

Done.

11) Subsection “Crystal structure shows the NSs core domain adopts a novel fold”, first paragraph, Table 1, and elsewhere: the "P" in P6422 should be italicized.

Done.

12) Figure 2 legend: It is stated that the protein chain is colored in a blue-to-red spectrum, but it would be more accurate to state that it is colored as a rainbow. A blue-to-red spectrum could just be blue-purple-red.

Done, also changed in the figure legend for Figure 4—figure supplement 1.

13) Introduction, first paragraph: "requiring urgent attention" was already mentioned in the same sentence and should be deleted.

Done.

14) Introduction, fourth paragraph: mention that this protein is an ORFan therefore having limited sequence similarity. Do the other Phleboviruses use these proteins and how related are they within the Bunya's?

This is addressed in the fourth paragraph of the Introduction and in the ninth paragraph of the Discussion. We expanded the relevant section in the Introduction to highlight sequence diversity and the limited knowledge of NSs protein functions and mechanisms.

15) Subsection “Crystal structure shows the NSs core domain adopts a novel fold”, second paragraph: it would be nice to mention the molecular mass of the protein somewhere.

Done.

16) Figure 2. Include the numbers and the N and C termini distinctions in the picture of the structure.

Done.